# Diagnosis of heart diseases: A fuzzy-logic-based approach

**Md. Liakot Ali** [ID]*, **Muhammad Sheikh Sadi** [ID], **Md. Osman Goni**

Institute of Information and Communication Technology (IICT), BUET, Dhaka, Bangladesh

* liakot@iict.buet.ac.bd

**Data Availability Statement:** All relevant data are within the paper and its Supporting Information files.

**Funding:** The author(s) received no specific funding for this work.

## Abstract

Cardiovascular diseases (CVD) also known as heart disease are now the leading cause of death in the world. This paper presents research for the design and creation of a fuzzy logic-based expert system for the prognosis and diagnosis of heart disease that is precise, economical, and effective. This system entails a fuzzification module, knowledge base, inference engine, and defuzzification module where seven attributes such as chest pain type, HbA1c (Haemoglobin A1c), HDL (high-density lipoprotein), LDL (low-density lipoprotein), heart rate, age, and blood pressure are considered as input to the system. With the aid of the available literature and extensive consultation with medical experts in this field, an enriched knowledge database has been created with a sufficient number of IF-THEN rules for the diagnosis of heart disease. The inference engine then activates the appropriate IF-THEN rule from the knowledge base and determines the output value using the appropriate defuzzification technique after the fuzzification module fuzzifies each input depending on the appropriate membership function. Moreover, the fusion of web-based technology makes it suitable and cost-effective for the prognosis of heart disease for a patient and then he can take his decision for addressing the problem based on the status of his heart. On the other hand, it can also assist a medical practitioner to reach a more accurate conclusion regarding the treatment of heart disease for a patient. The Mamdani inference method has been used to evaluate the results. The system is tested with the Cleveland dataset and cross-checked with the in-field dataset. Compared with the other existing expert systems, the proposed method performs 98.08% accurately and can make accurate decisions for diagnosing heart diseases.

## 1. Introduction

An expert system is a computer system that enables a human expert in making a decision using artificial-intelligence methods [1]. It is usually designed to solve complex problems by reasoning using a knowledge base Fig 1 shows the basic architecture of an expert system.

When an expert system uses fuzzy logic instead of bi-valued Boolean logic is known as a fuzzy expert system. A fuzzy expert system is a collection of fuzzy rules means knowledge base and membership functions that are used to manage data. A fuzzy logic-based expert system is

**Competing interests:** The authors have declared that no competing interests exist.

**Fig 1. Basic architecture of an expert system [1].**

widely used in the domain of medical diagnosis where there are numerous variables and uncertainties that affect the decision process thereby causing differences in the opinion of the practitioners [2–5]. In the medical application domain, there are usually imprecise conditions, and therefore fuzzy methods seem to be more suitable than crisp ones. Under such uncertainties to help physicians for making better decisions or for giving better treatment to the patient, computerized decision-supportive diagnostic tools are very much required nowadays.

It has been found in a survey by WHO, Cardiovascular Diseases (CVD) have come to be the leading cause of death globally [6]. Approximately 17.9 million people died from CVDs in 2016 which is 31% of all global deaths. A heart can be infected or affected by many factors and diseases which can result from different types of diseases of the human heart. There are many types of heart diseases that affect different parts of the organ and occur in different ways. These are Congenital Heart Disease, Arrhythmia, Coronary Artery Disease, Dilated Cardiomyopathy, Myocardial Infarction, Heart Failure, Hypertrophic Cardiomyopathy, Mitral Regurgitation, Mitral Valve Prolapse, Pulmonary Stenosis, Angina Pectoris, Endocarditis, Pericardial Disease, and Rheumatic Heart Disease, etc. People with heart diseases or who are at high cardiovascular risk need early detection and management using counseling and medicines, as appropriate. Most Heart diseases can be prevented by addressing behavioral risk factors. If an efficient and cost-effective fuzzy expert system is developed, then it could help the patients with the prognosis of heart disease and then can take their decision for addressing the problem based on the status of their heart. On the other hand, it can also assist a medical practitioner to reach a more accurate conclusion regarding the treatment of heart disease. Nowadays, web-based technology is a standard medium of exchanging information, and web-based communication technology benefits a person taking services anytime from anywhere globally [7, 8]. In this case, the fusion of web technology and the proposed expert system can provide an excellent platform that can assist a patient with the prognosis of heart disease at the same time can increase a heart specialist's confidence while diagnosing heart diseases. Hence, research is necessary to develop the said fuzzy logic-based expert system for the diagnosis of heart disease.

The following parts of this paper are organized as follows. In Section 2, related works in this area are described. In Section 3, input-output variables in the dataset for the proposed system, method of designing the fuzzy expert system that includes inference rules, fuzzification, and defuzzification methods are presented. In Section 4, the results are shown with the corresponding discussion, followed by a concluding remark in Section 5.

## 2. Related works

In the medical domain, accuracy is very important in case of diagnosis of any disease. In any expert system for medical diagnosis purposes, if a diagnosis does not offer acceptable accuracy, then it will never be acceptable to the patient as well as to the medical practitioner. Literature shows that fuzzy logic-based expert systems are frequently utilized in the diagnosis of disease where various factors influence the decision-making process and lead to variances in practitioners' opinions. It has already been shown the application of fuzzy logic and its effectiveness in the medical diagnosis of Ankylosing Spondylitis, anemia, dengue, thyroid, Alzheimer, Blood pressure, diabetes, and mental health, ontology-aided food and drug recommendation systems for a patient, etc. for identifying various disorders [5, 9–15]. Similarly, a lot of research is also going on to create an accurate, economical, and effective fuzzy logic-based expert system for diagnosing heart disease because the world health organization (WHO) reported that cardiovascular diseases (CVD) are now the main cause of death worldwide [5, 16–19]. Interestingly, the chance of mortality can be decreased by early diagnosis of heart diseases. However, adaptive algorithms are needed for earlier prediction. Some of them are presented in [20, 21]. A group of researchers in [22–31] works in the prediction of heart diseases. Another group of researchers in [32–39] focuses on the diagnosis of heart diseases. With a different number of input and output attributes, authors in [40–48] present different levels of accuracy percentages. The achieved accuracy is 63.24% [47] to 94.05% [44] based on various input and output features implemented in Mamdani inference systems. Literature [40] displays 94% accuracy using 44 rules in the fuzzy expert system with eleven inputs and one output. The accuracy is increased a bit more (94.05%) in [44] as the work introduced decision tree algorithm with the fuzzy expert system. In recent years, some research works [19, 49–52] presented further improvement of accuracy through an hybrid approach of fuzzy and neural network algorithm however hybridization makes the overall system complicated and computation intensive. Hence, this research will focus on development of solely fuzzy expert system for diagnosis of heart disease.

After reviewing all the literatures, it is observed that the expert systems are having limitations of acceptable accuracy and also none of them are reported as web based. Usually a fuzzy expert system heavily relies on its knowledge base to produce accurate result. So it is it is very much necessary and important to build a precise and accurate knowledge base having enough and sufficient number of inference rules in order to achieve acceptable accuracy in diagnosis of heart disease. So to develop the proposed expert system, all the IF-THEN rules available in the literature are collected and verified by medical experts in Bangladesh [53]. Then the knowledge base is further enriched adding all the possible IF-THEN rules through close consultation with the medical experts. The proposed Expert System has been simulated using the dataset of Cleveland Clinical Foundation in the (University of California, Irvine) UCI repository [54] and found the system's performance in terms of accuracy up to 98.08%. In addition, the system is tested in the laboratory to check the consistency as per rule, and the obtained results are very impressive. Finally, the results are compared with the existing research and it is observed that the proposed system outperforms these existing works in diagnosing heart diseases. In summary, the contributions of this work are:

- To develop a knowledge base enriched with all the possible number of IF-THEN rules which makes the proposed fuzzy expert system capable of producing diagnosis results with acceptable accuracy and outperform the existing fuzzy expert systems in this domain

- Fusion of web technology with the fuzzy expert system that makes the proposed fuzzy expert system a suitable and cost-effective platform for the prognosis of heart disease for a patient as well as for a doctor for diagnosis of heart diseases of a patient.

## 3. Background

The human heart is a muscular organ that controls the entire blood circulation system and circulates blood throughout the body. It is slightly to the left of the chest's center. It beats 60 times per minute on average for adults and circulates 5,000 gallons of blood through our bodies every 24 hours. Pushing blood transports oxygen and nutrient-rich blood to our tissues while also transporting the waste, including carbon dioxide.

### a. Data set

Our system is based on inference rules as a knowledge base created by the close consultation of Medical Experts. This system uses seven input fields/variables as input attributes for input and one output field/variable as an attribute for the result.

Input variables are Chest Pain Type, HbA1c, HDL, LDL, Heart Rate, Age, and Blood Pressure. The output variable refers to the presence of heart disease in the patient. The output variable indicates whether heart disease is there in the patient. It has universe of discourse valued from 0–4 (Healthy), 2–6 = Low Risk, 4–8 = Medium Risk, 6–10 = High Risk). HbA1c is used as Hemoglobin A1c, and systolic pressure is used as blood pressure. In this data, input variables are divided into some sections, and each section has a value. For in- stance, in this data, Chest Pain has four sections (1 = No Pain, 3 = Non Anginal, 5 = Atypical Angina Pain, and 7 = Typical Angina Pain), HbA1c has 3 sections (3–7 = Very Healthy, 6.5–9 = Healthy, and 8.5–14 = High), HDL has two sections (10–50 = Low and 40–80 Healthy), LDL has five sections (40–80 = Very Healthy, 70–110 = Healthy, 100–140 = High, 130–170 = Very High and 160–200 = Extra High), Heart Rate has three sections (40–70 = Very Healthy, 60–100 = Healthy and 90–160 = High), Age has four sections (80–120 = Very Old, 60–85 = Old, 40–65 = Mid and 20–45 = Young) and Blood Pressure has three sections (80–140 = Normal and 120–200 = High, and 180–240 = Very High). Expert's consultation was used to make these sections of input and output variables.

## 4. Materials and methods

The following sub-sections present the design of the membership functions, fuzzy system, fuzzy inference rules (knowledge base), and fuzzification and defuzzification techniques.

### a. Design of fuzzy expert system

Fig 2 shows a web-enabled fuzzy system for the diagnosis of heart diseases. It comprises of a control unit and a knowledge base unit. Seven functional modules integrate to a control unit of a web-based fuzzy expert system and have the following.

- **Inference Rule-Making Module**: In our expert system, the rules-making module offers a user interface as an input form so they may enter the IF and THEN portions of the rule. The input rule is then saved in a MySQL database using the standard IF...THEN text format. The IF-based logical expression that tries to emulate human-readable infix expression primarily employs the AND/OR logical operator to convert it to a postfix expression using a postfix algorithm. For usage throughout the inferencing process, the postfix expression of the IF portion is later recorded in the MySQL database.

  **The Input Process Module and the Update Process Module**: The purpose of the input process module is to make an input form available to the user or administrator so they can enter input values (in the proposed system) which are the symptoms of heart diseases. This module offers the capabilities of sending the user-provided input value to the Fuzzification

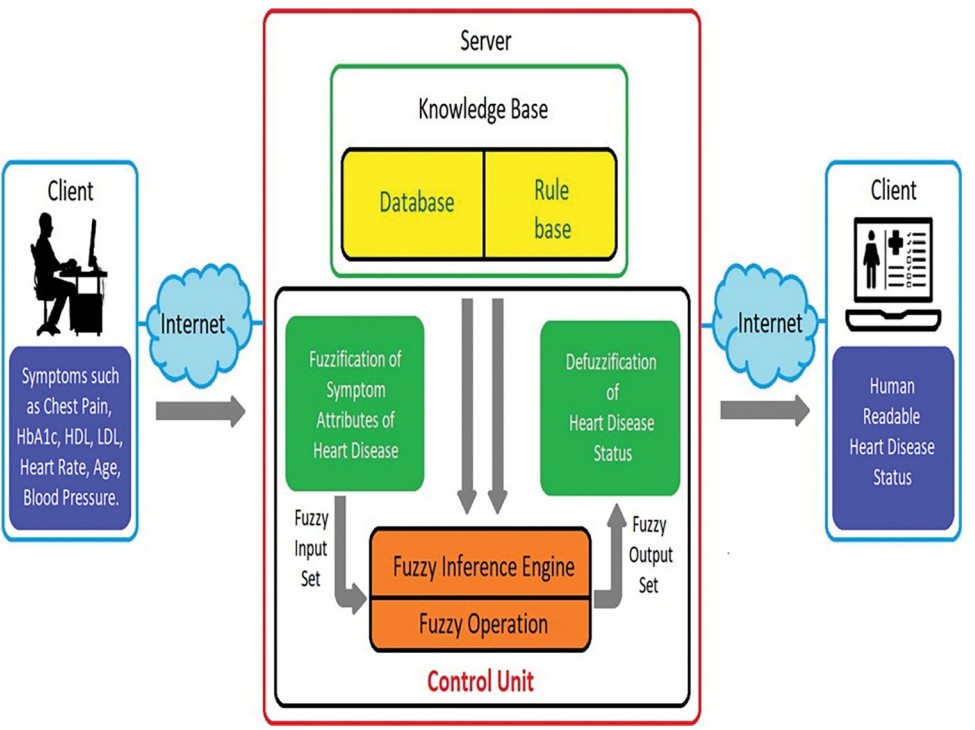

**Fig 2. Proposed design of the fuzzy expert system.**

Process Module for additional processing. This module displays the input or update form to the user in their web browser on their client computer. We can say that it serves as the system's user input form. The update process module's main job is to give the user or administrator an update form so they may enter the updated value (for the proposed system which are input variable, output variable, membership function and co-ordinate of membership function, etc.). It means the user/administrator is given user interface/form to enter/update information about the input variable, output variable, membership function and co-ordinate of membership function etc.

- **Fuzzification Process Module**: This module receives the input value—a numerical or verbal representation of the user's heart disease symptoms—through the input process module. The Fuzzification Process Module uses a defined fuzzifier—membership function—for the relevant input variables to transform the crisp input value into a corresponding fuzzy value. To establish what would affect the amount of these input variables to output variables, the proposed system will later use these fuzzified values in the associated fuzzy rules.

- **Fuzzy Inferencing and Operation Process Module**: In our system, the fuzzy Inferencing Process Module performs the functions of a fuzzy inference engine. The 'IF. . . THEN' rules kept in Knowledge Base (MySQL) and logical connector 'AND' are used by the fuzzy inferencing process module to produce a decision. This technique reduces the veracity of any rule or assertion to a matter of degree. The following is an example of a "IF. . . THEN" rule from our Knowledge Base: IF (Chest Pain: Non-Anginal; HbA1c: Normal; HDL: High; LDL: High; Heart Rate: Normal; Age: Middle; Blood Pressure: High) Low Risk THEN Status is Low Risk. Fuzzy Inferencing Process Module transforms the IF portion of the associated inference rule to the accumulated input variables' fuzzified value with the aid of the Fuzzy Operation Process Module. After that, the influence of these input variables on the

corresponding THEN part is determined. The proposed system has 4320 inference rules where every rule is formed with 7 input variables and the logical operator 'AND'. The logical operator 'AND' represents INTERSECTION which is obtaining the minimum value of all operands' fuzzy values in a statement. Consider an example: the membership function value of output G = MIN ($\mu_{chest\ pain}$, $\mu_{hba1c}$, $\mu_{hdl}$, $\mu_{ldl}$, $\mu_{heart\ rate}$, $\mu_{age}$, $\mu_{blood\ pressure}$). Hence, the system must do this operation to determine the MIN value of user-provided input variables that have had their values fuzzified. This is accomplished by the fuzzy operation process module of the proposed system employing a minimum value finding technique, with input coming from an array of the system's input variable. One or more fuzzy sets may be the result of the fuzzy inferencing process module, along with the output variable's matching membership values for each set. The Defuzzification Process Module will be supplied these fuzzy sets and membership values for additional processing.

- **Defuzzification Process Module**: The influence of THEN rule into output variable through Fuzzy Operations with all selected IF-THEN rules by doing MIN operations done by the Fuzzy Inferencing Process Module and Fuzzy Operation Module must be transformed into scalar or crisp values in the fuzzy expert system in order for the user to interpret the fuzzy output. The Fuzzy Inferencing Process Module generates a set of fuzzy sets and their accompanying membership values, which may be the input for the Defuzzification Process Module. Defuzzification is carried out by the module using three different methods: the aggregated area defuzzification, the weighted average defuzzification, and the mean of maxima defuzzification. The outcome is presented as the three defuzzification algorithms' average output of precise values.

- **Graphical Presentation Module**: The Graphical Presentation Module of our expert system uses jpGraph add-ons to exhibit the geometrical representation of the input, output, and generated results of fuzzy system graphically. Users can better grasp how input crisp values are transformed into fuzzy values and how these fuzzy values affect output variables thanks to this.

- **Knowledge Base Unit**: The foundation of a fuzzy expert system is a set of inferencing rules found in the fuzzy logic knowledge base. An excellent resource for storing the knowledge or inferencing rules of the fuzzy expert system is a relational database. The knowledge base is created in the MySQL database in our expert system. The coordinate points of all input and output variables are likewise saved and updated in the MySQL database. These rules are used by the fuzzy inferencing process module for inferencing from the MySQL database. The system was designed with the help of a relational database and knowledge base. The server houses the knowledge base and all the control programs. A client software, such as a web browser, can be used by the user to access and utilize the expert system. Users can access the system directly from the expert system server or via the internet.

## b. The flow chart of the proposed system

The flow chart of the proposed system (as shown in Fig 3) displays the proposed fuzzy expert system with a pre-designing tool for the system. It is detailed as follows.

- Step 1: Provide values for Chest Pain, HbA1c, HDL, LDL, Heart Rate, Age, and Blood Pressure individually.

- Step 2: Invalid value for any of Chest Pain, HbA1c, HDL, LDL, Heart Rate, Age, and Blood Pressure will compel to provide valid value within range of membership function.

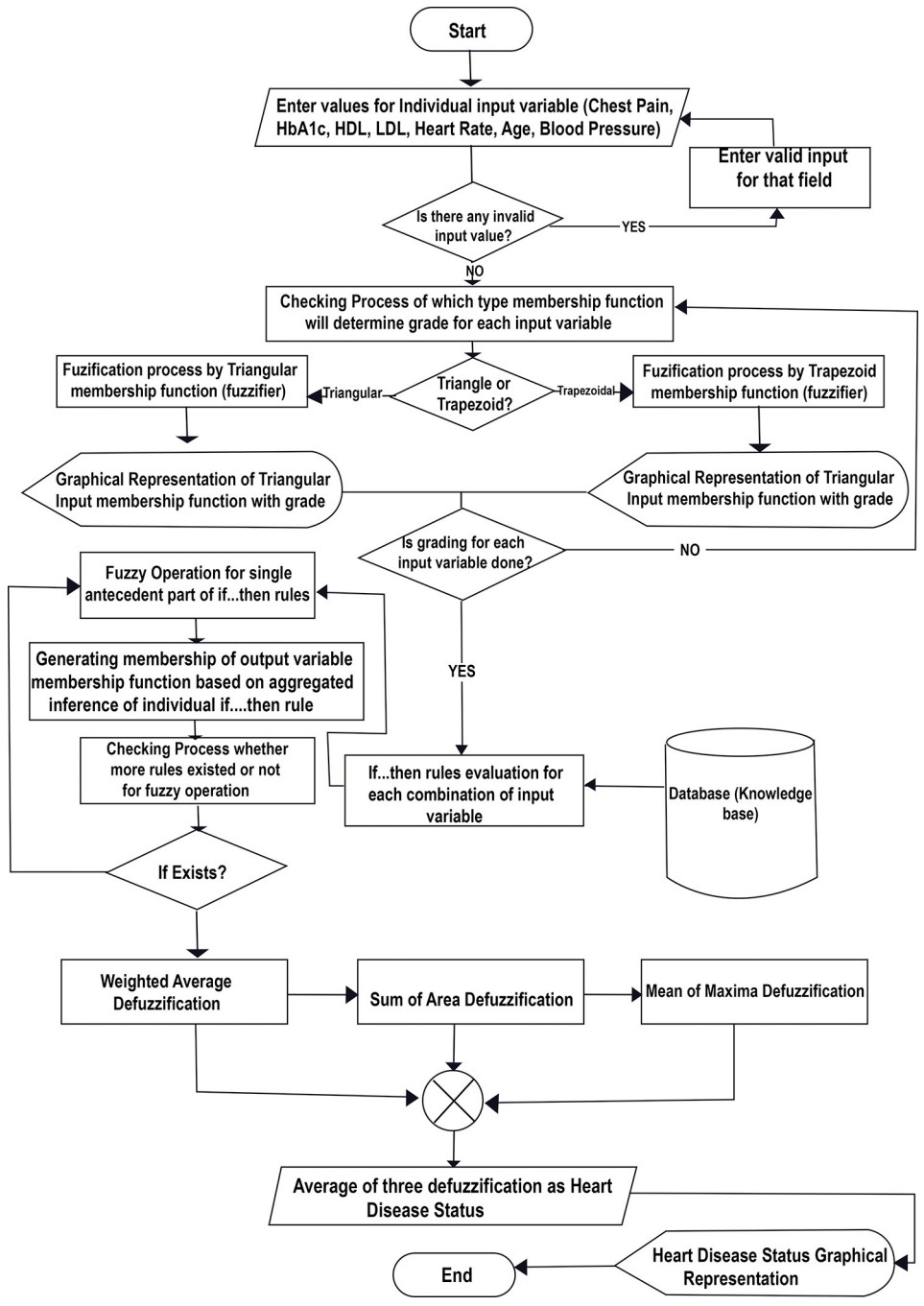

**Fig 3. System flow chart of the proposed expert system.**

- Step 3: The system will determine the membership function type (fuzzifier type) for provided value for each of the 7 attributes as input variables.

- Step 4: If the fuzzifier (membership function type) for the attribute (input variable) is a triangle, then triangular membership function does the fuzzification for finding membership value and will print the grade/membership value else if the fuzzifier (membership function type) is a trapezoid, then trapezoidal membership function does the fuzzification for finding membership value and will print the grade/membership value.

- Step 5: Repeat Step-3 and Step-4 one by one for all 7 attribute values to find their corresponding membership value.

- Step 6: Based on all possible membership values for 7 attributes corresponding inference rules (IF. THEN rules) will be selected from the knowledge base (MySQL Database).

- Step 7: Fuzzy operations for all selected antecedent part with If rules have to be done to produce corresponding membership/grade value of the output variable.

- Step 8: From the output variable after Fuzzy Operations, system will calculate the weighted average defuzzified value, the sum of the area defuzzified value, and the mean of the maxima defuzzified value.

- Step 9: System will find the arithmetic mean of the values obtained from the mentioned three defuzzification procedures in step 8.

- Step 10: System will display output variable with the outcome and display the arithmetic mean of the values obtained from the mentioned three defuzzification procedures as the crisp value for the status of heart disease.

## c. Designing the expert system based on fuzzy logic

The problem of dynamic behavior, uncertainties, and vagueness are suitable for using fuzzy logic. The selection of input and output variables, together with their sections depending on the domain of the universe of discourse, is the initial stage in creating a fuzzy expert system. There are seven attributes/symptoms as input variables and one output variable as heart disease status in the proposed system. The membership functions of all input and output variables are required to be designed. The level of membership of inputs or symptoms to fuzzy sets for input and output variables is determined by these membership functions. The input variables are shown as follows.

i. Chest pain: Chest pain symptom is an input variable that has four types of chest pain. For the sake of system testing, a value is defined in the proposed system for each form of chest pain. Every type of chest pain is a fuzzy set. Given that the patient only experiences one type of chest pain at a time, fuzzy sets are constructed in the form of crisp for this input variable and these do not overlap. Sorts of chest pain are 1 = No Pain, 3 = Non Anginal, 5 = Atypical, and 7 = Typical.

Chest Pain is input variable which has 4 chest pain types. We have defined a range in this system for each chest pain type that we use these values for system testing with consultation with physicians and by reviewing other related literature and medical articles. Each chest pain type is a fuzzy set. In this input variable, fuzzy sets do not have overlapping because the patient has just one chest pain type on time. Chest pain types are as follows- No Pain, Non Anginal, Atypical, Typical.

ii. HbA1c (Haemoglobin A1c): Three ranges of HbA1c are supported by HbA1c input variable. Each HbA1c level is a fuzzy set. HbA1c levels with their values have been shown as follows in Table 1. Membership functions of HbA1c is shown in Fig 4(A). Eq 1 shows these membership function expressions of HbA1c. Since HbA1c<3 does not have any impact on heart disease, membership value/grade is 0. When HbA1c is within 3–7 then there is no diabetic of the patience which is very healthy in term of heart disease. When It

**Table 1. The classification of HbA1c.**

| Input | Range | Fuzzy Sets |
|---|---|---|
| Hb1Ac | 3–7 | Very Healthy |
| | 6.5–9 | Healthy |
| | 8.5–14 | High |

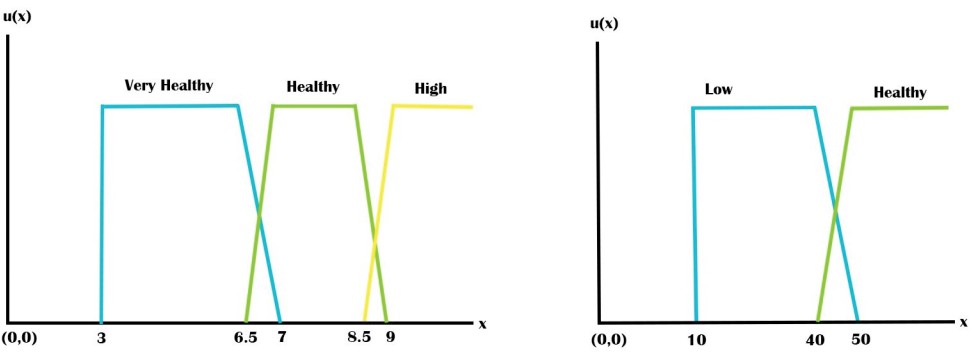

**Fig 4.** (a) Membership functions of HbA1c, (b) Membership functions of HDL.

crosses 6 then risk of heart diseases will be increased and for HbA1c $> 8.5$ risk factor for heart disease is highly increased. We have shown these in Table 1 and mathematically defined in Eq (1). Membership functions of HbA1c is constructed accordingly.

$$\mu_{Very\ Healthy}(x) = \begin{cases} 0, & if\ x < 3 \\ 1, & if\ 3 \le x \le 6.5 \\ \dfrac{7-x}{7-6.5}, & if\ 6.5 \le x \le 7 \\ 0, & if\ x > 7 \end{cases}$$

$$\mu_{Healthy}(x) = \begin{cases} 0, & if\ x < 6.5 \\ \dfrac{x-6.5}{7-6.5}, & if\ 6.5 \le x < 7 \\ 1, & if\ 7 \le x < 8.5 \\ \dfrac{9-x}{9-8.5}, & if\ 8.5 \le x \le 9 \\ 0, & if\ x > 9 \end{cases}$$

$$\mu_{High}(x) = \begin{cases} 0, & if\ x < 8.5 \\ \dfrac{x-8.5}{9-8.5}, & if\ 8.5 \le x < 9 \\ 1, & if\ x > 9 \end{cases} \qquad \text{Eq (1)}$$

 iii. HDL: Two ranges of HDL Cholesterol levels are supported by the HDL input variable. Table 2 and Fig 4(B) show information about it. Eq 2 shows these membership function expressions. If value of HDL is low that is not good for heart and with the increment of its value it becomes healthier for heart. So range of HDL 10–50 is lower heathy for heart disease and greater than 40 is healthier for heart. We have found value of HDL between 10–

**Table 2. The classification of HDL.**

| Input | Range | Fuzzy Sets |
|---|---|---|
| HDL | 10–50 | Low |
| | 40–80 | Healthy |

**Table 3. The classification of LDL, Heart Rate, Age and Blood Pressure.**

| Inputs | Range | Fuzzy Sets | Input Variable | Range | Fuzzy Sets |
|---|---|---|---|---|---|
| LDL | 40–80 | Very Healthy | Heart Rate | 40–70 | Very Healthy |
| | 70–110 | Healthy | | 60–100 | Healthy |
| | 100–140 | High | | 90–160 | High |
| | 130–170 | Very High | Age | 80–120 | Very Old |
| | 160–200 | Extra High | | 60–85 | Old |
| Blood Pressure | 40–70 | Medium | | 40–65 | Mid |
| | 60–100 | High | | 20–45 | Young |
| | 90–160 | Very high | | | |

50 is low healthy for heart and value between 40–80 is healthier for heart which are shown in Table 2 and membership functions of HDL cholesterol is constructed accordingly.

$$\mu_{Low}(\text{x}\} = \begin{cases} 0, & if\ x < 10 \\ 1, & if\ 10 \leq x \leq 40 \\ \dfrac{50 - x}{50 - 40}, & if\ 40 \leq x \leq 50 \\ 0, & if\ x > 50 \end{cases}$$

$$\mu_{Healthy}(\text{x}) = \begin{cases} 0, & if\ x < 40 \\ \dfrac{x - 40}{50 - 40}, & if\ 40 \leq x < 50 \\ 1, & if\ x > 50 \end{cases} \qquad \text{Eq (2)}$$

iv. LDL: Five ranges of LDL Cholesterol levels are supported by the LDL input variable. Table 3 and Fig 5(A) shows information about it. Eq 3 shows these membership function expressions. Lower value for LDL is good for heart and with the increment of value of LDL it becomes riskier for heart. We have found value of HDL between 40–80 is very healthy for heart, value between 70–110 is healthy, value between 100–140 is highly risky for heart, value between 130–170 is very highly risky for heart and value between 160–200 is extremely risky for heart which are shown in Table 3 and membership functions of LDL cholesterol is constructed accordingly.

$$\mu_{Very\ Healthy}(\text{x}) = \begin{cases} 0, & if\ x < 40 \\ 1, & if\ 40 \leq x \leq 70 \\ \dfrac{80 - x}{80 - 70}, & if\ 70 \leq x \leq 80 \\ 0, & if\ x > 80 \end{cases}$$

$$\mu_{Healthy}(\text{x}) = \begin{cases} 0, & if\ x < 70 \\ \dfrac{x - 70}{80 - 70}, & if\ 70 \leq x < 80 \\ 1, & if\ 80 \leq x < 100 \\ \dfrac{110 - x}{110 - 100}, & if\ 100 \leq x \leq 110 \\ 0, & if\ x > 110 \end{cases}$$

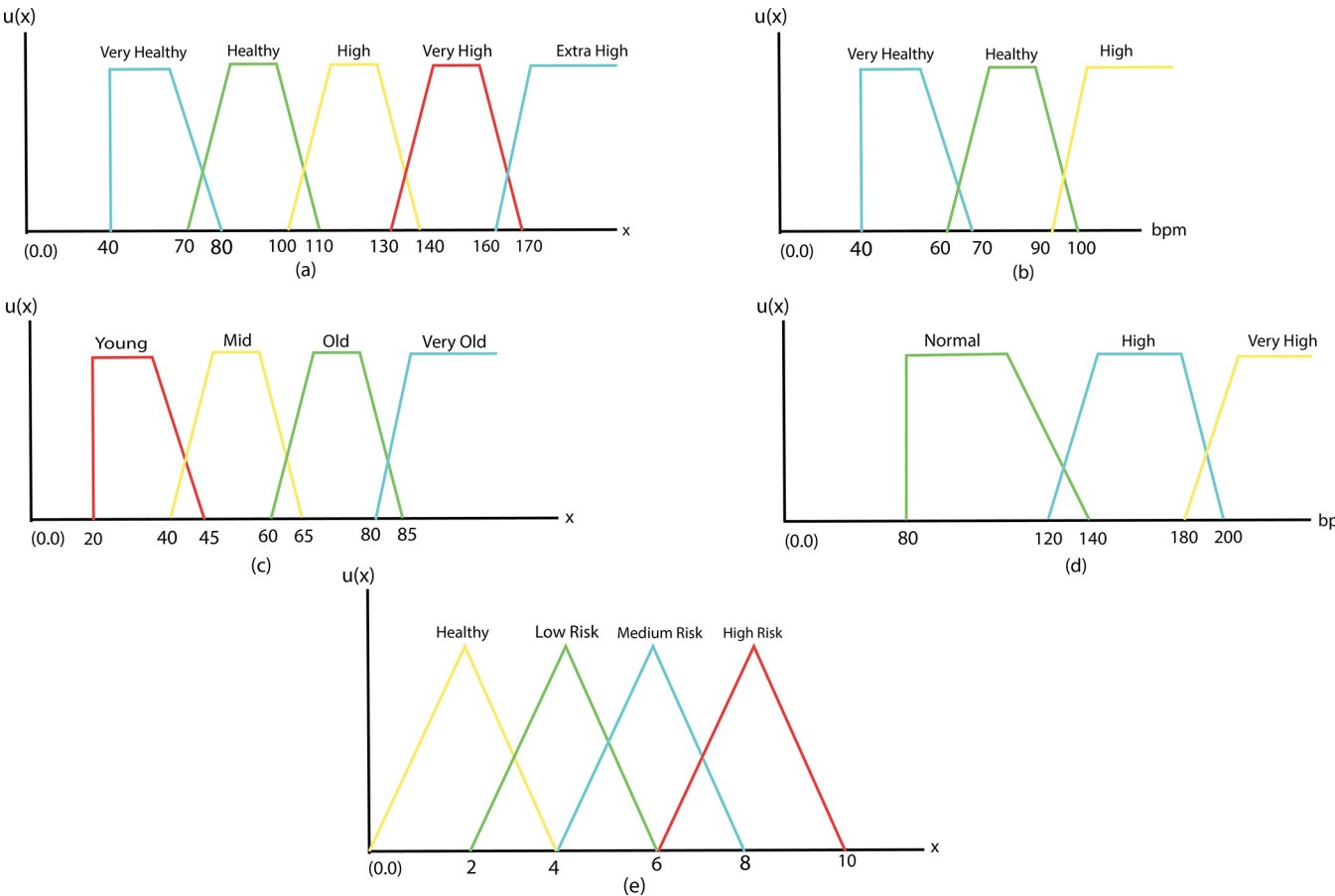

**Fig 5.** Membership functions of (a) LDL (b) heart rate, (c) age, (d) systolic blood pressure and (e) results.

$$\mu_{High}(x) = \begin{cases} 0, & if\ x < 100 \\ \dfrac{x-100}{110-100}, & if\ 100 \leq x < 110 \\ 1, & if\ 110 \leq x < 130 \\ \dfrac{140-x}{140-130}, & if\ 130 \leq x \leq 140 \\ 0, & if\ x > 140 \end{cases}$$

$$\mu_{Very\ High}(x) = \begin{cases} 0, & if\ x < 130 \\ \dfrac{x-130}{140-130}, & if\ 130 \leq x < 140 \\ 1, & if\ 140 \leq x < 160 \\ \dfrac{170-x}{170-160}, & if\ 160 \leq x \leq 170 \\ 0, & if\ x > 170 \end{cases}$$

$$\mu_{Extra\ High}(x) = \begin{cases} 0, & if\ x < 160 \\ \dfrac{x - 160}{170 - 160}, & if\ 160 \leq x < 170 \\ 1, & if\ x > 170 \end{cases} \qquad \text{Eq (3)}$$

v. Heart rate: Three ranges of Heart Rate levels are supported by Heart rate input variable. Table 3 and Fig 5(B) show information about it. Eq 4 shows these membership function expressions. Heart Rate between 40–70 bpm is healthier for heart, between 60–100 is healthy and greater than 100 it becomes riskier for heart which are shown in Table 3 and membership functions of HDL cholesterol is constructed accordingly.

$$\mu_{Very\ Healthy}(x) = \begin{cases} 0, & if\ x < 40 \\ 1, & if\ 40 \leq x \leq 60 \\ \dfrac{70 - x}{70 - 60}, & if\ 60 \leq x \leq 70 \\ 0, & if\ x > 70 \end{cases}$$

$$\mu_{Healthy}(x) = \begin{cases} 0, & if\ x < 60 \\ \dfrac{x - 60}{70 - 60}, & if\ 60 \leq x < 70 \\ 1, & if\ 70 \leq x < 90 \\ \dfrac{100 - x}{100 - 90}, & if\ 90 \leq x \leq 100 \\ 0, & if\ x > 100 \end{cases}$$

$$\mu_{High}(x) = \begin{cases} 0, & if\ x < 90 \\ \dfrac{x - 90}{100 - 90}, & if\ 90 \leq x < 100 \\ 1, & if\ x > 100 \end{cases} \qquad \text{Eq (4)}$$

vi. Age: Age input variable supports 4 ranges of Age. Table 3 and Fig 5(C) show information about it. The membership function expressions of Age (an input variable) are shown in Eq 5.

$$\mu_{Young}(x) = \begin{cases} 0, & if\ x < 20 \\ 1, & if\ 20 \leq x \leq 40 \\ \dfrac{45 - x}{45 - 40}, & if\ 40 \leq x \leq 45 \\ 0, & if\ x > 45 \end{cases}$$

$$\mu_{Mid}(x) = \begin{cases} 0, & if \ x < 40 \\ \dfrac{x-40}{45-40}, & if \ 40 \leq x < 45 \\ 1, & if \ 45 \leq x < 60 \\ \dfrac{65-x}{65-60}, & if \ 60 \leq x \leq 65 \\ 0, & if \ x > 65 \end{cases}$$

$$\mu_{Old}(x) = \begin{cases} 0, & if \ x < 60 \\ \dfrac{x-60}{65-60}, & if \ 60 \leq x < 65 \\ 1, & if \ 65 \leq x < 80 \\ \dfrac{85-x}{85-80}, & if \ 80 \leq x \leq 85 \\ 0, & if \ x > 85 \end{cases}$$

$$\mu_{Very \ Old}(x) = \begin{cases} 0, & if \ x < 80 \\ \dfrac{x-80}{85-80}, & if \ 80 \leq x < 85 \\ 1, & if \ x > 85 \end{cases} \qquad \text{Eq (5)}$$

vii. Blood pressure: Blood pressure input variable supports 3 ranges. Table 3 and Fig 5(D) show information about it. Eq 6 shows these membership function expressions.

$$\mu_{Normal}(x) = \begin{cases} 0, & if \ x < 80 \\ 1, & if \ 80 \leq x \leq 120 \\ \dfrac{140-x}{140-120}, & if \ 120 \leq x \leq 140 \\ 0, & if \ x > 140 \end{cases}$$

$$\mu_{High}(x) = \begin{cases} 0, & if \ x < 120 \\ \dfrac{x-120}{140-120}, & if \ 120 \leq x < 140 \\ 1, & if \ 140 \leq x < 180 \\ \dfrac{200-x}{200-180}, & if \ 180 \leq x \leq 200 \\ 0, & if \ x > 200 \end{cases}$$

$$\mu_{Very \ High}(x) = \begin{cases} 0, & if \ x < 180 \\ \dfrac{x-180}{200-180}, & if \ 180 \leq x < 200 \\ 1, & if \ x > 200 \end{cases} \qquad \text{Eq (6)}$$

**Table 4. The categorization of the status of heart diseases.**

| Output | The Range | The Fuzzy Sets |
|---|---|---|
| Result | 0–4 | Healthy |
| | 2–6 | Low Risk |
| | 4–8 | Medium Risk |
| | 6–10 | High Risk |

## 4.1 The status of heart disease

The output variable (heart disease status) refers whether heart disease is present or not in the patient and condition of heart can be healthy, in low risk, in medium risk or in high risk. The status of heart disease is determined based on seven input attributes/fields/symptoms in the system: Chest Pain, HbA1c, HDL, LDL, Heart Rate, Age, and Blood Pressure. Depending on the values of these attributes, heart disease may or may not be present. Heart disease status is an integer value and its value lies between 0 and 10, or absence. IF-THEN rule based on 7 attributes/symptoms have influence on this value range of output variable after doing inferencing and fuzzy operations and finally doing the defuzzification. The patient's chance of developing heart disease rises as the integer number rises. The output variable is divided into four (4) fuzzy sets (Healthy, Low Risk, Medium Risk, and High Risk). Table 4 shows these fuzzy sets with their ranges. Membership functions of **Healthy**, **LowRisk**, **MediumRisk,** and **HighRisk** fuzzy sets all are triangular. In Fig 5(e) these membership functions are shown.

## 4.2 The inference rules

A fuzzy logic system's core components or Inference rules (knowledge-based) are what determine the system's overall quality. The fuzzy system in question contains 4320 rules. Each rule's antecedent portion is divided into seven components. The idea of the expert and the outcomes of the laboratory are often the outcomes of this fuzzy system with its established rules. Table 5 provides a sample of a couple of inference rules. In this table, the first column shows the conditions of the inference rules (values of the attributes) and the second column shows the output (the status of the heart disease). For example, the first row of the first column mentions If

**Table 5. Sample of inference rules (knowledge based) of the system.**

| If Rule | Then Rule |
|---|---|
| 1. If (Chest Pain is Non Anginal AND LDL is High AND HDL is Healthy AND HbA1c is Healthy AND Heart Rate is Healthy AND Blood Pressure is Normal AND Age is Mid) | Status is Low Risk |
| 2. If (Chest Pain is Atypical AND LDL is XHigh AND HbA1c is Healthy AND HDL is Low AND Heart Rate is Very Healthy AND Blood Pressure is Normal AND Age is Mid) | Status is Low Risk |
| 3. If (Chest Pain is Typical AND HbA1c is Healthy AND LDL is XHigh AND HDL is Low AND Heart Rate is Very Healthy AND Blood Pressure is High AND Age is Mid) | Status is Medium Risk |
| 4. If (Chest Pain is Atypical AND HbA1c is Very Healthy AND LDL is Very Healthy AND HDL is Low AND Heart Rate is Very Healthy AND Blood Pressure is High AND Age is Mid) | Status is Medium Risk |
| 5. If (Chest Pain is Non Anginal AND HbA1c is High AND LDL is XHigh AND HDL is Low AND Heart Rate is Healthy AND Blood Pressure is High AND Age is Mid) | Status is High Risk |
| 6. If (HDL is Low AND Chest Pain is Typical AND HbA1c is Healthy AND LDL is XHigh AND Blood Pressure is High AND Heart Rate is Healthy AND Age is Mid) | Status is High Risk |
| 7. If (LDL is Healthy AND HbA1c is Very Healthy AND Chest Pain is No Pain AND HDL is Healthy AND Blood Pressure is Normal AND Heart Rate is Very Healthy AND Age is Young) | Status is Healthy |

*(Chest Pain is Non Anginal AND LDL is High AND HDL is Healthy AND HbA1c is Healthy AND Heart Rate is Healthy AND Blood Pressure is Normal AND Age is Mid)* as the condition of the inference rule. This condition detailed that the value of Chest Pain (attribute) is Non Anginal, the value of LDL is High, the value of HDL is Healthy, the value of HbA1c is Healthy, the value of Heart Rate is Healthy, the value of Blood Pressure is Normal, and the value of Age is Mid. The first row of the second column shows the output (status) of the heart disease for these conditions (in this case, the status is low risk). Similarly, other rows of this table are formed.

The approach described in the literature is used to construct 4320 inference rules in our fuzzy expert system [40]. Following that, we modified the Cleveland database's detection rules for heart disease in the UCI repository. The rules given by our local medical expert are cross-checked against any remaining rules that are not connected to the Cleveland database.

A statistical link between the individual membership function of the input variable and output variable (heart disease status) is obtained using the methodology described in the literature [40]. In both the input and output variables, the middle point value of every linguistic term membership function is used. The Pearson's statistical coefficient of correlation between two variables or qualities can be expressed as shown in Eq 7 [55] if $x_i$, and $y_i$ are the individual sample points indexed by i of n sample size and $x_1$, $y_1$ are the mean of the sample of all $x_i$, $y_i$.

$$r_{xy} = \frac{\sum_{i=1}^{n}(x_i * y_i) - n * \bar{x} * \bar{y}}{\sqrt{\left(\sum_{i=1}^{n} x_i^2 - n * \bar{x}^2\right)} * \sqrt{\left(\sum_{i=1}^{n} y_i^2 - n * \bar{y}^2\right)}} \qquad \text{Eq (7)}$$

n = sample size

$x_i$, $y_i$ are the individual sample points indexed with i.

$\bar{x} = \frac{1}{2}\sum_{i=1}^{n} x_i$, Mean of sample of all $x_i$

$\bar{y} = \frac{1}{2}\sum_{i=1}^{n} y_i$, Mean of sample of all $y_i$

We consider the sample size, n = 4; therefore, the statistical coefficient of correlation between chest pain and heart disease status using Eq 7 is 1. Similarly, the correlation coefficients of all other six can be calculated. Fig 6 displays the Pearson correlation of heart disease with seven factors. Chest pain and heart disease are perfectly positively correlated, as indicated by the correlation coefficient of r = +1 between the two variables. The correlation metric is indicated by the various values of r. Given that HDL and heart disease have a perfect negative correlation (r = -1), we can conclude that there is negative relationship between HDL and heart disease.

## 4.3 Fuzzification and defuzzification

For this fuzzy expert system, Mamdani's approach is used as the inference mechanism. This system uses the AND logical operator for the antecedent part of all inference rules to combine all logical combinations of input variables, producing a conclusion as the output. Each input variable's membership from crisp values and its matching membership section determines the membership or degree of each rule.

Let $g_x$ is the membership value for each input variable so, $g_{chestpain} = chestpain(x)$, $g_{hba1c} = HbA1c(x)$, $g_{hdl} = hdl(x)$, $g_{ldl} = ldl(x)$, $g_{heartrate} = heartrate(x)$, $g_{age} = age(x)$, $g_{bloodpressure} = bloodpressure(x)$.

For the aggregation of Inference Rules/Antecedent, MIN for AND logical operator is used i.e., G = MIN ($g_{chest\,pain}$, $g_{hba1c}$, $g_{hdl}$, $g_{ldl}$, $g_{heart\,rate}$, $g_{age}$, $g_{blood\,pressure}$).

In this fuzzy expert system, an average output in crisp value from defuzzified values of a weighted average (WA), a sum of area (SA), and mean of maxima (MM) defuzzification

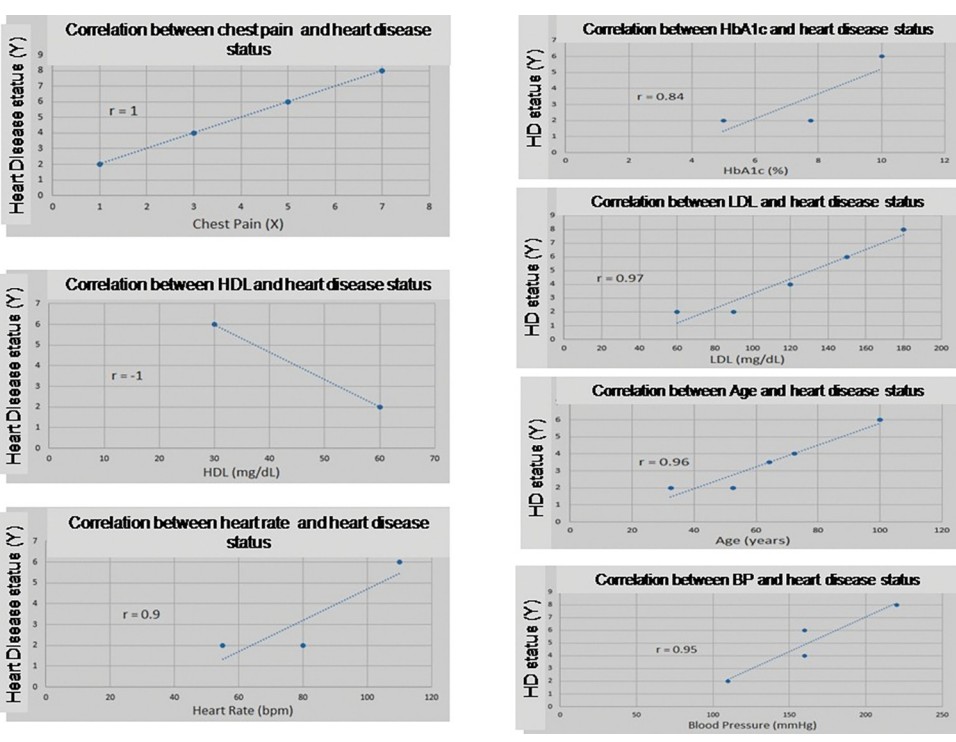

**Fig 6. Pearson correlation of heart disease with HDL, HbA1c, LDL, heart rate, age, and blood pressure.**

method is used as follows.

$$Output\ Value = \frac{WA + SA + MM}{3} \qquad \text{Eq (8)}$$

## 4.4 Design and implementations

Our proposed system environment is implemented using a use case diagram that presents the user interactions with it, an entity-relationship diagram (ERD) of database schema, and a software prototype. The proposed system is simulated in a custom web environment. The results can be immediately accessed using any computing device in this system. The MD5 hash function is used to hash sensitive information, such as passwords of user accounts.

## 4.5 Use case diagram

A use case diagram (as shown in Fig 7) shows interactions between the user and the system with the proposed system. This system has three types of actors: super admin, admin, and user. The system will display an error if the user already exists when the super admin creates a user. All actors can access the system through user verification, and if any unauthorized user information or credentials are entered, the system will display an error message. The problem name and necessary UOM were generated and modified by the super admin and admin, and errors could be present. In order to create rules connected to a problem, they can also create and update the relevant input and output variables, their accompanying membership functions, and their coordinate positions. For the existing irrelevant data, errors are produced. If irrelevant data is provided, all actors can observe the results of fuzzification, fuzzy inferencing, defuzzification, and errors. Additionally, the problem name, rules, UOM, any input/output variable, associated membership function, and their coordinate locations can be deleted by both super admin and admin.

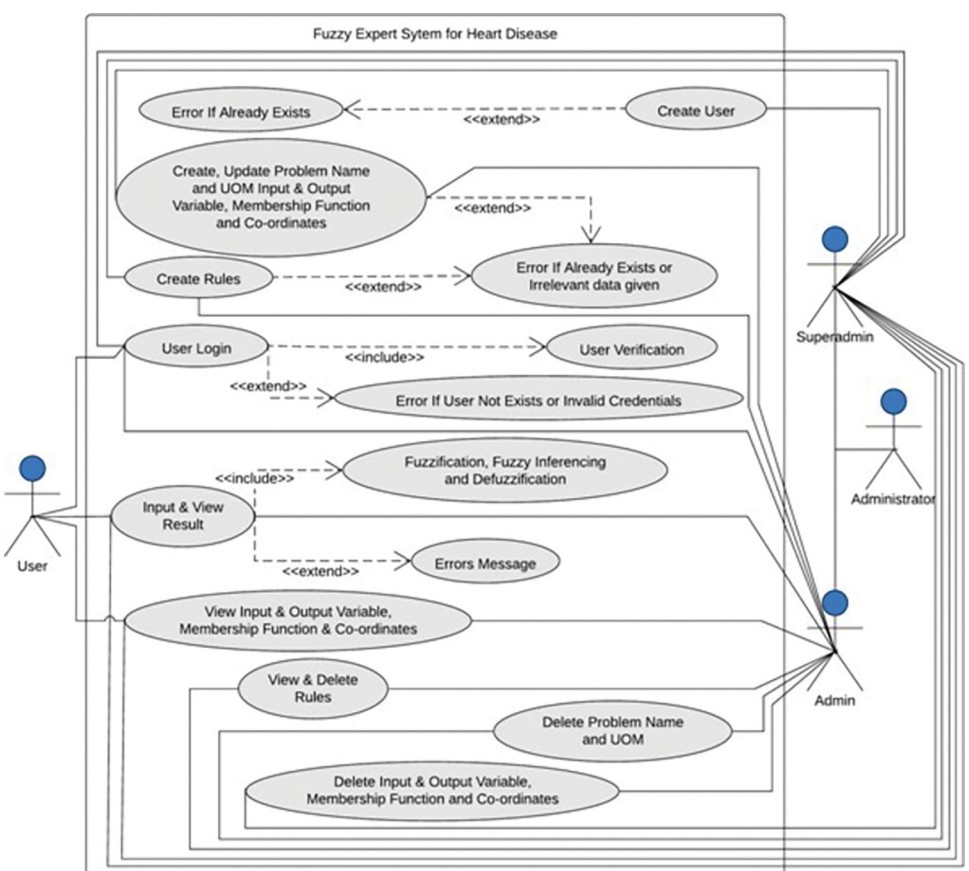

**Fig 7. Use case diagram for the heart disease detection system.**

## 4.6 Entity relationship diagram (ERD)

The proposed ERD diagram illustrates the cardinal relationships between the entities employed in our expert system, such as one-to-one (1:1), one-to-many (1:M), and many-to-many (M:N). User login, output variable table, universe of discourse info, input variable table, input membership function name, problem name, rules, input membership function coordinate point, output membership function name, and output membership function coordinate point are the entities in this work. A user may generate one or more issues. Therefore, there is a one-to-many relationship between the user login and the problem name (1:M). Fig 8 displays ER in tabular format. a representation of our expert system database.

Heart disease is a disorder with numerous characteristics or signs. Therefore, the relationship between the problem name and the input variable table is one to many (1:M). For the universe of discourse, an input variable has one or more input memberships set within its bounds. Thus, there is also a one-to-many (1:M) relationship between the input variable table and the input membership function name relations. Table 6 shows input parameters for the system testing and corresponding results.

There are coordinate points corresponding to each membership function. Therefore, the relation between input and coordinate points is one to many (1:M). Similar relationships exist between the problem and the output variable, the output variable and the output membership function, and the output membership function and the output membership function coordinate points. All of these relationships have a one to many relationship (1:M). Our expert

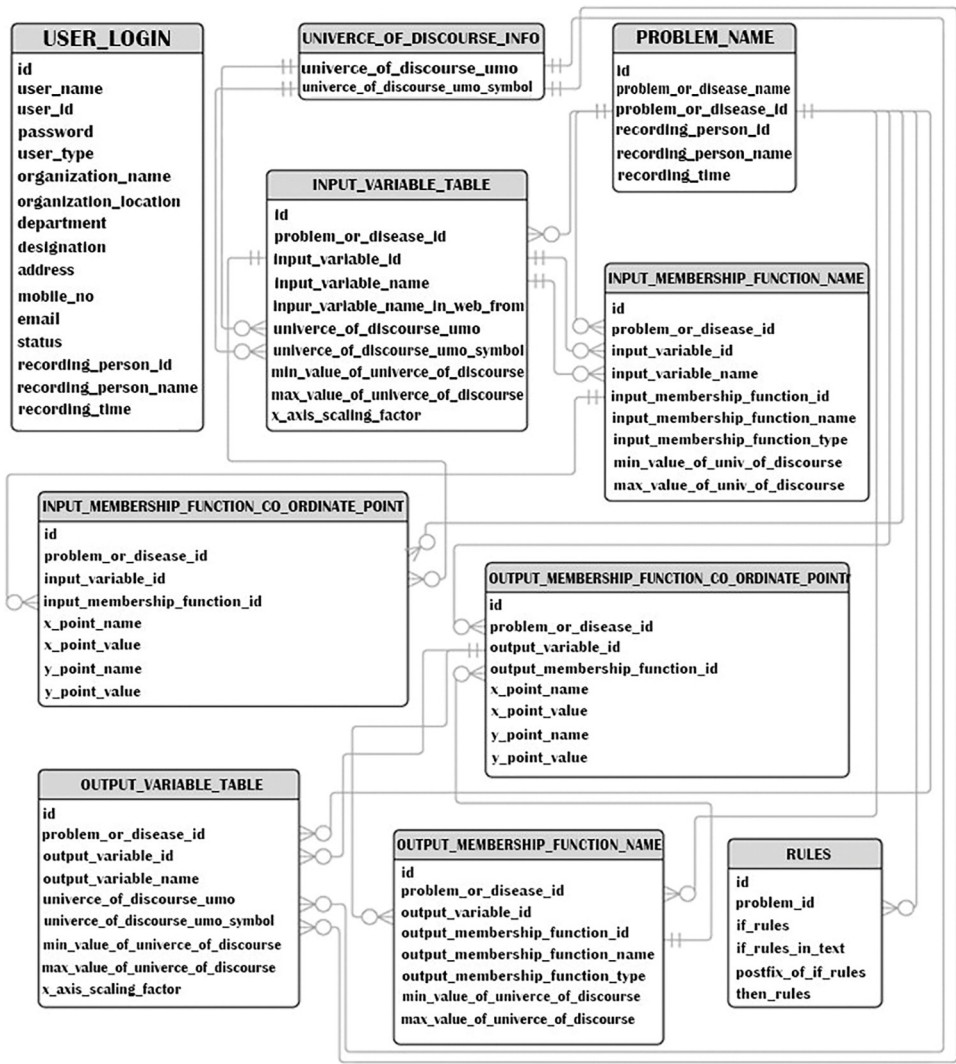

**Fig 8. ER diagram for the database of the expert system.**

system has several rules for the heart disease problem. As a result, the relationship between the problem name and the rules relation is one to many (1:M). In our expert system, relationships are not overly numerous. One regulation, for instance, cannot encompass all the issues. Specific rules or a rule can only apply to one problem.

**Materials.** The proposed fuzzy expert system is developed using the following technologies.

**Table 6. Sample input parameters for system testing and corresponding results.**

| Chest Pain | HbA1c | HDL | LDL |
|---|---|---|---|
| 5 | 10% | 30 | 141 |
| (A typical) | (Healthy) | (Low) | (Very High) |
| Heart rate | Age | Blood Pressure | Result of System |
| 60 | 59 | 126 | 6.44 |
| (Very Healthy) | (Mid) | ((Normal High) | (Medium Risk) |

**HTML.**   HTML as one of the core technologies in use on the World Wide Web and serves as the backbone of all webpages is used as front-end language in our system which will make our system visible/readable/intractable to user through web browser in any computing device.

**CSS.**   Cascading Style Sheets (CSS) as a style-sheet language that is paired with HTML is used in our system for making web pages stylish or for making format of web pages of our system.

**Bootstrap.**   For making web application development faster and easier, Bootstrap as a front-end web development framework is used in our system for typography, forms, buttons, tables, navigation, modules and images in our system. Also Bootstrap helps our system to be responsive in various computing devices of various screen size and resolution.

**Scripting languages.**   For dynamic webpages, we added more advanced client-side and server-side scripting. JavaScript, jQuery and AJAX are the two most commonly used client-side scripting language are used in our system. For the requirement of Control Program Unit of our system, a core program is required which has ability to handle all modules of Control Program Unit and can make interaction with Database for the inferencing rules. PHP as server-side scripting language is used in our system. Programming code of Input and Update Process Module, Rules Making Module, Fuzzification Process Module, Fuzzy Operation Process Module, Fuzzy Inferencing Process Module, Defuzzification Process Module and Graphical Presentation Module etc. were written using PHP language in our system.

**MySQL database.**   In our system MySQL as a relational database management system (RDBMS) is used as storage system of all defined/designed fuzzy inference rules. MySQL is used here as container of Knowledge Base. It the key component for Knowledge Base Unit of our program.

## 5. Results and discussions

The system has been tested using known input parameters to ensure the system's proper functionality. Table 6 shows sample input parameters for system testing and its corresponding outcome. It is found that our system provides the desired result, as shown graphically in Fig 9. The Fuzzy profile of the input-output parameters is presented in Fig 9(a)-9(h) where Y-Axis shows the fuzzy membership function of each parameter against its value in the X-Axis.

The combination of symptoms and input variables satisfy the two following inference rules.

Rule 1: If the chest pain is atypical, the HbA1c level is high, the HDL level is low, the LDL level is very high, the heart rate is very healthy, the age is mid-range, and the blood pressure is normal, Then the status is medium risk.

Rule 2: If the chest pain is atypical, the HbA1c level is high, the HDL level is low, the LDL level is very high, the heart rate is very healthy, the age is mid-range, and the blood pressure is high, Then the status is high risk.

For Rule 1: Medium Risk was identified as a heart disease state by our expert system with a membership value of 0.7. Medium Risk = MIN (1,1,1,1,1,1,0.7) = 0.7. For Rule 2: High Risk was identified as a heart disease state by our expert system with a membership value of 0.3 High Risk = MIN (1,1,1,1,1,1,0.3) = 0.3. After defuzzification (average of weighted average, sum of area, and mean of maxima) was applied, the output variable resulted in a crisp value of 6.44, which denotes a medium risk for heart disease.

We used 260 datasets of input variables from the Cleveland Clinic Foundation database to do benchmark testing on the functionality of our system. It is available at the UCI machine learning Repository. Only five input variables combinations were classified wrongly by our expert system, and 255 are classified accurately, which are in the Cleveland database. That results in 5 variable input combinations wrongly classified by our system but close to the

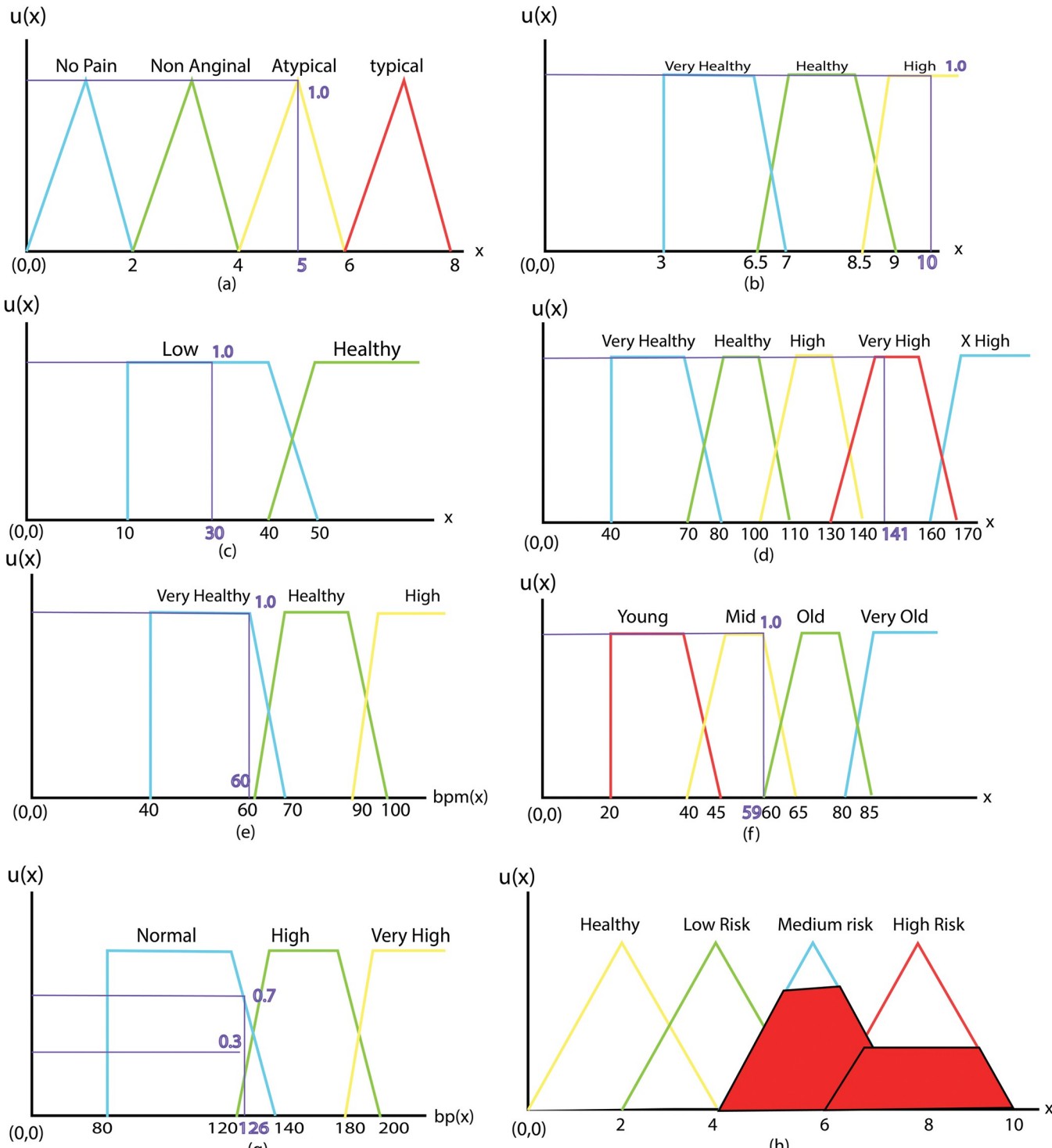

**Fig 9.** Membership functions of (a) Chest Pain (b) HbA1c, (c) HDL Cholesterol, (d) LDL Cholesterol (e) Heart Rate, (f) Age, (g) Systolic Blood Pressure, and (h) results.

**Table 7. The comparison between the performance of the proposed system and that of existing work.**

| Test done by | Published year | Method type | Accuracy |
|---|---|---|---|
| Literature [16] | 2016 | Diagnosis: type-2 Fuzzy | 73.78% |
| Literature [17] | 2015 | Prediction: hybrid genetic fuzzy | 86.00% |
| Literature [31] | 2014 | Diagnosis: fuzzy model | 88.79% |
| Literature [39] | 2010 | Diagnosis: fuzzy model | 94.00% |
| Literature [44] | 2016 | Prediction: fuzzy model | 94.05% |
| Literature [47] | 2019 | Diagnosis: fuzzy model | 91.00% |
| Literature [51] | 2020 | Diagnosis: fuzzy model | 96.60% |
| Literature [18] | 2021 | Diagnosis: fuzzy model | 94.55% |
| Literature [48] | 2017 | Diagnosis: fuzzy neural networks | 97.78% |
| Our Expert System | Current | Diagnosis fuzzy Model | 98.08% |

classification value. It might be overcome if we combine our present five heart disease detection input factors with one or more input variables acquired from the Cleveland database.

Classification Accuracy: 260 symptom/attribute combinations from the UCI repository's Cleveland database were used. The following formula is used to get the percentages of classification accuracy. Hence, in this case, Classification Accuracy = (Correct Classified Patterns)/(Total Patterns)*100% = 255/260*100% = 98.08%

Table 7 shows the comparison of the performance of the proposed system with those of similar existing systems offered by other researchers. A graphical representation of the comparison of the performance is shown in Fig 10. It shows that our approach outperforms all the existing systems.

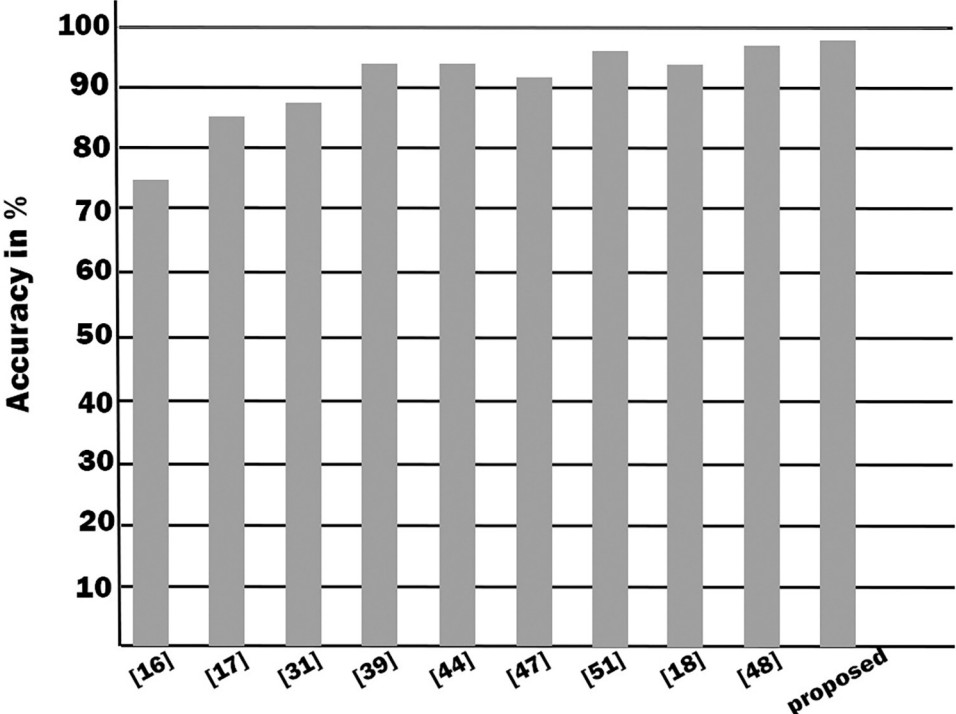

**Fig 10. Comparison with other research works.**

This proposed fuzzy rule-based expert system efficiently predicts heart disease and replaces manual efforts. It is more proficient and faster, more accurate than manual work. This system involves two major phases, one which performs classification and diagnosis, the other one that detects the rate of risks of heart diseases. For this system, the Mamdani inference system is used. Using 260 symptoms and features combined from the UCI repository's Cleveland database, the percentage of classification accuracy was found 98.08% by our system whereas Literatures [17,18,32] have an accuracy of 73.78%, 86%, and 88.79% respectively. The results show that the proposed system outperforms the existing works in diagnosing heart disease.

## 6. Conclusions

This paper proposes an effective fuzzy logic-based expert system for the diagnosis of heart diseases. The literature and extensive contact with medical professionals in this field was used to establish a sufficient number of inference rules in the proposed system. The proposed system can assist a heart specialist in the diagnosis of heart disease accurately and reliably and increase his level of confidence. It has been tested in the laboratory and found that it is working as per the developed inference rule. The simulated results with the Cleveland data sets deliver the accuracy of the system up to 98.08%. The results show that the proposed system outperforms the existing works in diagnosing heart disease.

The proposed system has not been tested on sufficient number of real patients in the field that means in a hospital or in a medical chamber. Hence, in the future, this paper can be enhanced by experimenting with a sufficient number of real patients in the field which means the experiment can be accomplished on the real-patients in a hospital or in a medical chamber. Moreover, the neuro-fuzzy approach may be incorporated into the proposed system to improve its functionality.

## Supporting information

**S1 File.**
(ZIP)

## Acknowledgments

This paper is based on the post graduate thesis work of Md. Osman Goni where Md. Liakot Ali was the supervisor. Authors are grateful to IICT, BUET for allowing us to use its all kinds of laboratory facilities to conduct this work. We acknowledge the support of ICT Division of Ministry of Posts, Telecommunications and Information Technology, Bangladesh for providing ICT Postdoctoral fellowship to Muhammad Sheikh Sadi who has contributed in preparing this paper.

## Author Contributions

**Conceptualization:** Md. Liakot Ali, Md. Osman Goni.

**Formal analysis:** Md. Osman Goni.

**Investigation:** Md. Osman Goni.

**Methodology:** Md. Osman Goni.

**Resources:** Md. Liakot Ali.

**Software:** Md. Liakot Ali, Md. Osman Goni.

**Supervision:** Md. Liakot Ali.

**Validation:** Md. Liakot Ali.

**Visualization:** Md. Liakot Ali.

**Writing – original draft:** Md. Liakot Ali, Md. Osman Goni.

**Writing – review & editing:** Md. Liakot Ali, Muhammad Sheikh Sadi.

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
