## [Decision Letter · Decision Letter 0]

17 Oct 2022

PONE-D-22-26721A Web-based Fuzzy Expert System for Diagnosis of Heart Diseases  PLOS ONE

Dear Dr. Md. Liakot Ali,

Thank you for submitting your manuscript to PLOS ONE. After careful consideration, we feel that it has merit but does not fully meet PLOS ONE’s publication criteria as it currently stands. Therefore, we invite you to submit a revised version of the manuscript that addresses the points raised during the review process.

Please submit your revised manuscript by Dec 01 2022 11:59PM. If you will need more time than this to complete your revisions, please reply to this message or contact the journal office at plosone@plos.org. Please include the following items when submitting your revised manuscript:A rebuttal letter that responds to each point raised by the academic editor and reviewer(s). You should upload this letter as a separate file labeled 'Response to Reviewers'.A marked-up copy of your manuscript that highlights changes made to the original version. You should upload this as a separate file labeled 'Revised Manuscript with Track Changes'.An unmarked version of your revised paper without tracked changes. You should upload this as a separate file labeled 'Manuscript'.

We look forward to receiving your revised manuscript.

Kind regards,

Muhammad Fazal Ijaz

Academic Editor

PLOS ONE

Journal Requirements:

3. Please upload a new copy of Figure 8 as the detail is not clear. Please follow the link for more information: https://blogs.plos.org/plos/2019/06/looking-good-tips-for-creating-your-plos-figures-graphics/" https://blogs.plos.org/plos/2019/06/looking-good-tips-for-creating-your-plos-figures-graphics/

Reviewers' comments:

Reviewer's Responses to Questions

**Comments to the Author**

1. Is the manuscript technically sound, and do the data support the conclusions?

Reviewer #1: Yes

Reviewer #2: No

Reviewer #3: Yes

2. Has the statistical analysis been performed appropriately and rigorously? 

Reviewer #1: Yes

Reviewer #2: No

Reviewer #3: Yes

3. Have the authors made all data underlying the findings in their manuscript fully available?

Reviewer #1: Yes

Reviewer #2: No

Reviewer #3: Yes

4. Is the manuscript presented in an intelligible fashion and written in standard English?

Reviewer #1: Yes

Reviewer #2: Yes

Reviewer #3: No

5. Review Comments to the Author

Reviewer #1: Manuscript holds great interest as it needs revision as follows.

-Figure quality needs to be improved

-Dataset attribute requires to represent in paper.

-The statement "Our system is based on inference rules as a knowledge base created by the close consultation of --Medical Experts." need to be justified by valid references of medical experts to select inputs.

-Justify your approach is a success amidst popular Deep learning and Machine Learning Models. Contrast it.

Authors should study the following paper for better understanding

https://www.mdpi.com/1424-8220/21/23/8095

Reviewer #2: 1. Abstract need to be elaborated where proposed solution must be explained in detail

2. Change the title of paper

3. Explain contribution of the work try to add one more paragraph in introduction

4. The flow chart steps need to be well explained more in detail

5. Explain chest pain in detail

6. Designing the expert system based on fuzzy logic need detail explanation

7. Equation 1,2 and 3 are not well explained in the text

8. The status of heart disease is poorly explained

9. Explain table 5 in detail

10. Add future directions separate section after conclusion

11. Try to check overall typo errors there are many errors in writing

Reviewer #3: In this paper, authors described the design and creation of a web-based fuzzy expert system for the diagnosis of heart disease. However, there are some limitations that must be addressed as follows.

1. The abstract is not attractive. Some sentences in abstract should be modified to make it more attractive for readers.

2. In Introduction section, it is difficult to understand the novelty of the presented research work. This section should be modified carefully. In addition, the main contribution should be presented correctly. Include more details in bullets.

3. The related work section should be included, and the healthcare monitoring systems should be discussed as follows. ‘A smart healthcare monitoring system for heart disease prediction based on ensemble deep learning and feature fusion’, ‘An intelligent healthcare monitoring framework using wearable sensors and social networking data’, ‘Automatic detection of Alzheimer’s disease progression: An efficient information fusion approach with heterogeneous ensemble classifiers’, ‘ANAF-IoMT: A Novel Architectural Framework for IoMT-Enabled Smart Healthcare System by Enhancing Security Based on RECC-VC’, Alzheimer’s disease progression detection model based on an early fusion of cost-effective multimodal data’ and ’ Type-2 fuzzy ontology–aided recommendation systems for IoT–based healthcare’. In addition, In the last lines of Literature review, highlight what overall technical gaps are observed in existing works that led to the design of proposed methodology.

4. Captions of the Figures and tables not self-explanatory. These captions should be self-explanatory, and clearly explaining the figure. Extend the description of the mentioned figures and tables to make them self-explanatory.

5. All the fuzzy sets should be represented in one table. See the above type-2 fuzzy ontology.

6. All figures are blurred, their quality should be improved.

7. The inference rules are not correct. Authors should confirm these rules again.

8. The authors should discuss the above existing work in table 7.

9. The conclusion section should be revised. In addition, the future work should be included.

6. PLOS authors have the option to publish the peer review history of their article (what does this mean?). If published, this will include your full peer review and any attached files.

Reviewer #1: **Yes: **Dr Jana Shafi

Reviewer #2: No

Reviewer #3: No

---

## [Author Response · Author response to Decision Letter 0]

15 May 2023

Please see the attached "Response to Reviewers' Comments" File.

---

## [Decision Letter · Decision Letter 1]

26 Jun 2023

PONE-D-22-26721R1Diagnosis of Heart Diseases: A Fuzzy-Logic-based ApproachPLOS ONE

Dear Dr. Ali,

Thank you for submitting your manuscript to PLOS ONE. After careful consideration, we feel that it has merit but does not fully meet PLOS ONE’s publication criteria as it currently stands. Therefore, we invite you to submit a revised version of the manuscript that addresses the points raised during the review process.

We look forward to receiving your revised manuscript.

Kind regards,

Tien V.T. Nguyen

Academic Editor

PLOS ONE

Journal Requirements:

Additional Editor Comments:

Dear Dr. Md. Liakot Ali:

Thank you for providing the point-to-point response.

I am writing to inform you of the decision regarding your manuscript titled "Diagnosis of Heart Diseases: A Fuzzy-Logic-based Approach" (PONE-D-22-26721), which has undergone a rigorous peer-review process. Congratulations on the quality and significance of your research, and I appreciate the time and effort you have invested in this study.

After carefully considering your manuscript and the insightful feedback from the reviewers, I am pleased to inform you that your manuscript has been assessed as having significant potential for publication in PLOS ONE. However, to ensure that the manuscript meets the highest standards and addresses the concerns raised by the reviewers, I have decided to classify this submission as requiring a MINOR REVISION.

I would like to emphasize that this decision is not a reflection of the quality of your work but rather an opportunity further to enhance the clarity and impact of your research. The comments and suggestions provided by the reviewers have been instrumental in shaping the revision process and will aid in strengthening the manuscript. I would like to express my gratitude for their valuable insights, which will undoubtedly contribute to the overall improvement of your research.

In light of these comments, please address each reviewer's concerns in your revised manuscript. Please provide clear and detailed responses to each comment, and indicate the changes made in the revised manuscript to facilitate the review process. Additionally, I would appreciate it if you could submit a separate document listing the revisions made and addressing each reviewer's comments point-by-point.

The revised manuscript should be submitted to the submission system by the mentioned deadline for revision submission. If you require an extension for any valid reason, please get in touch with me as soon as possible to discuss the possibility of an extension. Please note that failure to submit the revised manuscript by the given deadline may result in a reassessment of the manuscript's status.

If you encounter any difficulties during the revision process or have any questions, please do not hesitate to contact me. I am here to provide assistance and guidance to ensure a smooth and efficient revision.

Once again, I commend you on your valuable contribution to the field and the potential impact of your research. I look forward to receiving your revised manuscript and working with you to bring your study to its full potential.

Thank you for your attention to this matter.

Yours sincerely,

Tien V.T. Nguyen

Reviewers' comments:

Reviewer's Responses to Questions

**Comments to the Author**

1. If the authors have adequately addressed your comments raised in a previous round of review and you feel that this manuscript is now acceptable for publication, you may indicate that here to bypass the “Comments to the Author” section, enter your conflict of interest statement in the “Confidential to Editor” section, and submit your "Accept" recommendation.

Reviewer #3: (No Response)

Reviewer #4: (No Response)

Reviewer #5: All comments have been addressed

2. Is the manuscript technically sound, and do the data support the conclusions?

Reviewer #3: (No Response)

Reviewer #4: Yes

Reviewer #5: Partly

3. Has the statistical analysis been performed appropriately and rigorously? 

Reviewer #3: (No Response)

Reviewer #4: Yes

Reviewer #5: Yes

4. Have the authors made all data underlying the findings in their manuscript fully available?

Reviewer #3: (No Response)

Reviewer #4: Yes

Reviewer #5: Yes

5. Is the manuscript presented in an intelligible fashion and written in standard English?

Reviewer #3: (No Response)

Reviewer #4: Yes

Reviewer #5: No

6. Review Comments to the Author

Reviewer #3: The authors have addressed my all comments. I have no further comments. Therefore, this paper can be accepted in its present form.

Reviewer #4: I understand that this manuscript was reviewed and revised based on previous reviewers suggestions. However, in my opinion, the content of the manuscript is quite unbalanced, that is, the introduction is quite long while the results of the research are relatively short. In addition, I think the authors should move the discussion of the findings to the Conclusion. Likewise, the authors can move the contents of Section 7, Future Directions, into the end of the Conclusion & Discussion.

Reviewer #5: Dear Editor and Author:

I have carefully reviewed the manuscript " Diagnosis of Heart Diseases: A Fuzzy-Logic-based Approach" submitted to PLOS ONE. I appreciate the authors' efforts in presenting their research on designing and creating a fuzzy logic-based expert system for the prognosis and diagnosis of heart disease. However, I have several concerns regarding the manuscript's structure, content, and clarity that need to be addressed before it can be considered for publication.

(for more detail, please read the attachment)

7. PLOS authors have the option to publish the peer review history of their article (what does this mean?). If published, this will include your full peer review and any attached files.

Reviewer #3: No

Reviewer #4: **Yes: **Kittisak JERMSITTIPARSERT

Reviewer #5: No

---

## [Author Response · Author response to Decision Letter 1]

12 Sep 2023

We would like to thank the anonymous associate editor and reviewers for their careful reading and constructive feedback on the previous draft entitled “Diagnosis of Heart Diseases: A Fuzzy-Logic-based Approach”. We have carefully considered their comments in preparing our revision, which has resulted in a manuscript that is clearer, more compelling, and broader.

Comments of Reviewer:

....................................................................................................................................................................

Comment #1: Regarding Structure and Organization:

The manuscript lacks a clear and well-structured organization, which makes it difficult to follow the flow of information. I suggest the authors reorganize the manuscript into sections such as Introduction,

Materials and Methods, Results, Discussion, and Conclusion to provide a logical study progression.

Response: Thank you for your insightful observation. It is certainly a valuable comment to enrich the manuscript. The current manuscript is reorganized and updated as per suggestion. For details, please see the manuscript. It is highlighted by yellow color.

Comment #2: Regarding Methodology:

The description of the proposed fuzzy logic-based expert system is insufficient. The authors briefly mention the system's components, such as the fuzzification module, knowledge base, inference engine, and defuzzification module, but fail to explain how these components work together comprehensively. It is crucial to provide a detailed account of the fuzzy logic techniques employed, including the membership functions, rules, and defuzzification technique, to ensure the readers can fully understand the methodology.

Response: Thank you for your comment. I would like to mention politely that the working procedure of the proposed system is shown by the flowchart of the system (with detailed illustration) on page 7.

Comment #3: Regarding Dataset and Evaluation:

The authors mention testing the system using the Cleveland dataset and cross-checking with an in-field

dataset but fail to provide specific details about these datasets. It is essential to provide information on

these datasets' characteristics, size, and source. Additionally, the authors should explain the selection

criteria used for the datasets and provide a thorough analysis of the evaluation results to support their

claim of achieving 98.08% accuracy.

Response: Thank you for your insightful observation. For testing the proposed system, a benchmark dataset of Cleveland Clinical Foundation in the (University of California, Irvine) UCI repository has been used. It is mentioned on page 4 of the manuscript. Details of the benchmark datasets can be found in the reference # [54]. We have mentioned cross-checking with an in-field dataset. The in-field data set is an example data input of a patient from a doctor of a hospital as mentioned in reference # [53]. It is shown in Table 6 of the revised manuscript on page 23.

Comment #4: Regarding Clarity of Figures:

The figures in the manuscript are blurry and illegible, making it challenging to comprehend the information presented. I strongly recommend the authors improve the quality and resolution of the figures to ensure readability. Additionally, providing detailed captions and labels for each figure would enhance their clarity and help readers understand the visual information effectively.

Response: Thank you for your comment. I would like to mention politely that the figure is system generated and it has been enhanced as best as possible as per comment of the previous reviewers. Moreover, the quality of the given figures is tested and verified by the automated tool “PACE” as recommended by the journal.

Comment #5: In summary, this manuscript requires substantial revisions to improve its structure, clarify its methodology, enhance the quality of figures, and emphasize the significance of the findings. Addressing these concerns will strengthen the manuscript and ensure its suitability for publication in PLOS ONE.

Response: Thank you for your insightful observation and comments. I would like to mention politely that necessary measures have been taken as per comments where it has been possible.

I appreciate your precious comment that these would strengthen the manuscript for its publication.

---

## [Decision Letter · Decision Letter 2]

6 Oct 2023

Diagnosis of Heart Diseases: A Fuzzy-Logic-based Approach

PONE-D-22-26721R2

Dear Dr. Ali,

We’re pleased to inform you that your manuscript has been judged scientifically suitable for publication and will be formally accepted for publication once it meets all outstanding technical requirements.

Kind regards,

Tien V.T. Nguyen

Academic Editor

PLOS ONE

Additional Editor Comments (optional):

Dear Dr. Md. Liakot Ali:

Thank you for submitting your work to Plos ONE.

I am writing to you - the corresponding author of the manuscript " Diagnosis of Heart Diseases: A Fuzzy-Logic-based Approach," which is submitted to PLOS ONE for consideration.

The reviewers' comments were invaluable in shaping our decision, and I would like to express my sincere gratitude for their time and effort. We received all thoughtful and constructive reviews; the reviewers recommended "Accept" after 3 rounds of review.

Please read the detailed comments and suggestions in the reviewer's comments section.

I have considered the reviewers' suggestions and made **Accept** for publication after minor revisions, as mentioned in the reviewers' comments and suggestions section.

Thank you for your time and attention.

Yours sincerely,

Reviewers' comments:

Reviewer's Responses to Questions

**Comments to the Author**

1. If the authors have adequately addressed your comments raised in a previous round of review and you feel that this manuscript is now acceptable for publication, you may indicate that here to bypass the “Comments to the Author” section, enter your conflict of interest statement in the “Confidential to Editor” section, and submit your "Accept" recommendation.

Reviewer #4: All comments have been addressed

Reviewer #5: All comments have been addressed

2. Is the manuscript technically sound, and do the data support the conclusions?

Reviewer #4: Yes

Reviewer #5: Yes

3. Has the statistical analysis been performed appropriately and rigorously? 

Reviewer #4: Yes

Reviewer #5: Yes

4. Have the authors made all data underlying the findings in their manuscript fully available?

Reviewer #4: Yes

Reviewer #5: Yes

5. Is the manuscript presented in an intelligible fashion and written in standard English?

Reviewer #4: Yes

Reviewer #5: Yes

6. Review Comments to the Author

Reviewer #4: For the paper "Diagnosis of Heart Diseases: A Fuzzy-Logic-based Approach", in my opinion, all comments have bee addressed.

Reviewer #5: Dear Editor and Authors:

Thank you for providing the point-to-point response.

The authors have carefully and patiently corrected and answered the comments and questions. The manuscript looks perfect now, and the reviewer suggests it be accepted for publication in this journal.

Please feel free to contact me if you have further requests or concerns.

Thank you for reading.

7. PLOS authors have the option to publish the peer review history of their article (what does this mean?). If published, this will include your full peer review and any attached files.

Reviewer #4: **Yes: **Kittisak JERMSITTIPARSERT

Reviewer #5: No

---

## [Editor Report · Acceptance letter]

31 Oct 2023

PONE-D-22-26721R2 

Diagnosis of Heart Diseases: A Fuzzy-Logic-based Approach 

Dear Dr. Ali:

I'm pleased to inform you that your manuscript has been deemed suitable for publication in PLOS ONE. Congratulations! Your manuscript is now with our production department. 

Kind regards, 

on behalf of

Dr. Tien V.T. Nguyen 

Academic Editor

PLOS ONE